# Aerobic Exercise Training Improves Calcium Handling and Cardiac Function in Rats with Heart Failure Resulting from Aortic Stenosis

**DOI:** 10.3390/ijms241512306

**Published:** 2023-08-01

**Authors:** Vítor Loureiro da Silva, Gustavo Augusto Ferreira Mota, Sérgio Luiz Borges de Souza, Dijon Henrique Salomé de Campos, Alexandre Barroso Melo, Danielle Fernandes Vileigas, Priscila Murucci Coelho, Paula Grippa Sant’Ana, Carlos Padovani, Ana Paula Lima-Leopoldo, Silméia Garcia Zanati Bazan, André Soares Leopoldo, Antonio Carlos Cicogna

**Affiliations:** 1Department of Internal Medicine, Botucatu Medical School, São Paulo State University (UNESP), Botucatu 18618-687, Brazil; gustavo.mota@unesp.br (G.A.F.M.); sergio.souza@unesp.br (S.L.B.d.S.); dijon.campos@unesp.br (D.H.S.d.C.); dani.vileigas@iq.usp.br (D.F.V.); paulagrippa@yahoo.com.br (P.G.S.); sgzanati@fmb.unesp.br (S.G.Z.B.); ac.cicogna@unesp.br (A.C.C.); 2Department of Sports, Federal University of Espirito Santo, Vitória 29075-910, Brazil; alexandre--barroso@hotmail.com (A.B.M.); priscilamurucci@hotmail.com (P.M.C.); ana.leopoldo@ufes.br (A.P.L.-L.); andresoaresleopoldo@gmail.com (A.S.L.); 3Department of Biostatistics, Institute of Biosciences, São Paulo State University (UNESP), Botucatu 18618-689, Brazil; padovani@ibb.unesp.br

**Keywords:** exercise training, aortic stenosis, heart failure, isolated papillary muscle, isolated cardiomyocyte, calcium handling

## Abstract

Aerobic exercise training (AET) has been used to manage heart disease. AET may totally or partially restore the activity and/or expression of proteins that regulate calcium (Ca^2+^) handling, optimize intracellular Ca^2+^ flow, and attenuate cardiac functional impairment in failing hearts. However, the literature presents conflicting data regarding the effects of AET on Ca^2+^ transit and cardiac function in rats with heart failure resulting from aortic stenosis (AoS). This study aimed to evaluate the impact of AET on Ca^2+^ handling and cardiac function in rats with heart failure due to AoS. Wistar rats were distributed into two groups: control (Sham; *n* = 61) and aortic stenosis (AoS; *n* = 44). After 18 weeks, the groups were redistributed into: non-exposed to exercise training (Sham, *n* = 28 and AoS, *n* = 22) and trained (Sham-ET, *n* = 33 and AoS-ET, *n* = 22) for 10 weeks. Treadmill exercise training was performed with a velocity equivalent to the lactate threshold. The cardiac function was analyzed by echocardiogram, isolated papillary muscles, and isolated cardiomyocytes. During assays of isolated papillary muscles and isolated cardiomyocytes, the Ca^2+^ concentrations were evaluated. The expression of regulatory proteins for diastolic Ca^2+^ was assessed via Western Blot. AET attenuated the diastolic dysfunction and improved the systolic function. AoS-ET animals presented an enhanced response to post-rest contraction and SERCA2a and L-type Ca^2+^ channel blockage compared to the AoS. Furthermore, AET was able to improve aspects of the mechanical function and the responsiveness of the myofilaments to the Ca^2+^ of the AoS-ET animals. AoS animals presented an alteration in the protein expression of SERCA2a and NCX, and AET restored SERCA2a and NCX levels near normal values. Therefore, AET increased SERCA2a activity and myofilament responsiveness to Ca^2+^ and improved the cellular Ca^2+^ influx mechanism, attenuating cardiac dysfunction at cellular, tissue, and chamber levels in animals with AoS and heart failure.

## 1. Introduction

The practice of exercise training plays an important role in: (1) improvement of the physical fitness of practitioners, which is essential to optimizing the performance of daily activities [1,2]; (2) disease prevention, making it unlikely that the active individual in youth develops risk factors for chronic diseases [3]; (3) mitigation of the effects of possible diseases related to aging [1,4,5,6,7]; and (4) attenuation of the consequences of current diseases [1,5,6,8,9,10].

Lower cardiorespiratory fitness and worse functional capacity and quality of life are common features of heart failure [11,12,13]. Aerobic exercise training (AET) is an important therapeutic tool in order to improve cardiac function and, consequently, the functionality of individuals with heart failure.

Experimental studies evaluating rats with heart failure show that AET improves physical conditioning, functional capacity, and cardiac function by enhancing several physiological mechanisms, such as calcium (Ca^2+^) handling, one of the main factors in positive cardiac adaptation [14,15,16,17,18,19,20,21,22,23,24,25,26]. Specifically, in cardiac dysfunction resulting from different models of experimental aortic stenosis (AoS), the literature presents a lack of studies and divergence of results regarding the participation of Ca^2+^ handling and its regulatory proteins in improving cardiac function by AET [14,26]. In heart-diseased pigs with AoS and preserved ejection fraction, low-intensity aerobic exercise training improved the functional characteristics of the isolated cardiomyocytes, and this positive adaptation is related to the attenuation of the damage to the Ca^2+^ handling mechanism [14]. The authors observed an increase in the sarcoplasmic reticulum calcium ATPase (SERCA2a)/phospholamban (PLB) ratio, PLB phosphorylated in serine 16, and the Na^+^/Ca^2+^ exchanger (NCX), in addition to a decrease in diastolic Ca^2+^. In contrast, van Deel et al. [26] found that voluntary exercise, incorporated immediately after surgery, did not restore SERCA2a and RyR levels in mice with severe AoS.

Thus, this study aimed to evaluate the effects of AET on Ca^2+^ handling and cardiac function in rats with heart failure due to AoS. We hypothesize that AET improves cardiac function through adaptations intrinsic to the myocardium, involving modulation of components and the functioning of Ca^2+^ handling. Our findings showed that animals with AoS developed heart failure, and AET was able to generate positive adaptations in physical fitness, the heart, and mechanisms of Ca^2+^ sarcoplasmic reuptake and cellular influx in the myocardium.

## 2. Results

### 2.1. AET Improves Systolic and Diastolic Cardiac Function, and Signs of Heart Failure in Animals with AoS

Echocardiograms were performed before and after AET, i.e., after 18 and 28 weeks of the experimental protocol. The first evaluation showed that 18 weeks of AoS caused important structural and functional cardiac changes (Table 1), such as predominantly concentric cardiac hypertrophy (↑PWDT, ↑ISDT, and ↑RWT), left atrium dilation (↑LA/Ao), and diastolic (↑E wave, ↑E/A, ↑E/E′, and EDT) and systolic dysfunction (↓PWSV). The AoS-ET group also showed decreased FS and EF. In the second evaluation, after adding ten weeks to the experimental protocol, the animals submitted to AoS remained with structural and functional alterations similar to those in the first evaluation (Table 1). In addition to the losses verified in the first moment, the AoS group presented ↑LVDD and LVSD and ↓A′, E′, and FS compared to the Sham group. AoS-ET animals presented positive cardiac adaptations after the AET protocol in relation to AoS (Table 1), such as lower magnitude atrial dilatation (↓LA/Ao) and improved systolic (↑PWSV and EF) and diastolic function (↑E′, A′, and EDT, and ↓E wave, E/A and E/E′). Furthermore, clinical and pathological signs were evaluated in the animals at the end of 28 weeks. The AoS group had a higher frequency of ascites, pleural effusion, left atrial thrombi, hepatic congestion, and tachypnea compared to the AoS-ET group. Furthermore, the survival rate did not differ between the AoS and AoS-ET groups.

### 2.2. AoS Impairs the Rate of Tension Development and Resting Tension, Whereas AET Is Effective in Recovering Resting Tension in AoS Animals

An isolated papillary muscle study showed that AoS impaired the myocardial contractile and relaxation functions, as observed by a reduced maximum rate of tension development and increased rest tension (Table 2). AET enhanced muscle relaxation function, whereas rest tension declined in the AoS-ET group compared to AoS animals.

### 2.3. AoS Deteriorates SERCA2a Function, While AET Attenuates the Loss of SERCA2a Functionality in AoS-ET Animals

Figure 1A–C presents the papillary muscle response percentage to post-rest contraction (10, 30, and 60 s). The AoS group showed functional impairment after the pauses performed in the three variables studied in relation to Sham, except for the 10 s pause for the variables −dT/dt and +dT/dt. The AoS-ET group demonstrated worse performance compared to the Sham-ET group after the 30 and 60 s pauses for the variables DT and +dT/dt and after the 60 s pause for −dT/dt. Both findings from the groups with heart disease suggest that AoS reduced the Ca^2+^ recapture potential of the sarcoplasmic reticulum. On the other hand, AoS-ET animals had enhanced performance compared to AoS animals after the 60 s pause for the DT and −dT/dt variables, which suggests positive adequacy in Ca^2+^ recapture. Figure 1D–F shows papillary muscle response percentages to elevation of extracellular Ca^2+^ (1.5, 2.5, and 3.5 mM). There were no changes in the functional response between the experimental groups for all variables analyzed.

Figure 2A–C expresses papillary muscle responses to SERCA2a inhibition and increasing Ca^2+^ concentration. After SERCA2a inhibition, no differences were observed in the functional response among the four experimental groups for the variables studied.

Figure 2D–F presents papillary muscle responses to L-type Ca^2+^ channel inhibition and increasing Ca^2+^ concentrations. The AoS group showed significantly greater deterioration in performance than the Sham group for all variables assessed at practically all times, except for the 0.5 mM Ca^2+^ concentration for TD, +dT/dt, and −dT/dt (Figure 2D–F), suggesting impaired function of L-type Ca^2+^ channels 28 weeks after surgery. AET attenuated these deleterious effects of AoS since the papillary muscles from AoS-ET animals expressed similar performance to the Sham animals during the maneuver, as well as lower DT, +dT/dt, and −dT/dt depression when compared to AoS animals at concentrations of 1.5, 2.0, and 3.5 mM of Ca^2+^ (Figure 2D–F). Additionally, there was no difference between the Sham-ET and AoS-ET groups for all variables at all evaluated moments.

In summary, the post-rest contraction maneuver showed that AoS animals did not respond inotropically to the stimulation return after pausing the electrical stimulus; an inotropic response would be expected because SERCA2a continues recapturing Ca^2+^ during the stimulus rest. There are three hypothetical causes for this physiological impairment: (1) functional loss of SERCA2a; (2) impairment of mechanisms for Ca^2+^ entry into the cell and/or Ca^2+^ release by RyR; and (3) loss of myofilament sensitivity to Ca^2+^. AoS animals presented significant functional loss compared to Sham and AoS-ET groups after blocking of L-type calcium channels by diltiazem, also showing impairment of Ca^2+^ channel function in these animals. In this context, none of the hypotheses mentioned above as explanations for the worst performance in the post-rest contraction maneuver could be completely discarded.

Overall, the experimental maneuvers demonstrated that AoS animals present deterioration of SERCA2a and L-type Ca^2+^ channels and that AET may be an important therapeutic tool to attenuate these deleterious effects of severe pressure overload.

### 2.4. AET Improved the Shortening and Relaxation Function in AoS Animals at a Cellular Level

Figure 3A–F describes cardiomyocyte mechanical function. The AoS group showed impairment in the following variables compared to the Sham group: fractional shortening, shortening maximum velocity, time to 50% shortening, and time to 50% relaxation. Furthermore, the AoS-ET group presented worse shortening of maximum velocity than Sham-ET. AET was efficient in improving the shortening and relaxation capacities of cardiomyocytes. The AoS-ET group showed a higher velocity of shortening and relaxation, a shorter time to 50% relaxation, and enhanced fractional shortening than the AoS group.

### 2.5. AET Improves the Responsiveness of Myofilaments to Ca^2+^ in the Isolated Cardiomyocyte Experiment

Figure 4A–H summarizes Ca^2+^ handling functioning after 28 weeks of the experimental protocol, which included ten weeks of AET. AoS animals presented an increase in the time to Ca^2+^ peak and time to 50% of Ca^2+^ decay in comparison to Sham animals, whereas AoS-ET animals presented a decrease in the time to Ca^2+^ peak and time to 50% of Ca^2+^ decay in comparison to the AoS group. AET enhanced the sensitivity of myofilaments to Ca^2+^ in animals submitted to AoS since the AoS-ET group had a lower value for Systolic Ca^2+^/FS and Systolic Ca^2+^/SMV ratios than the AoS group.

### 2.6. AoS Causes Changes in Ca^2+^ Handling Proteins, Which Are Restored in Animals Submitted to AET

Figure 5A–I shows the protein expressions that regulate myocardial Ca^2+^ handling. AoS caused an increase in SERCA2a and NCX when comparing the Sham and AoS groups (Figure 5B,G) and in PLBthr17 between Sham-ET and AoS-ET (Figure 5I). On the other hand, AET prevented the rise in NCX and SERCA2a in the AoS-ET group, which presented a lower amount of these proteins than the AoS group (Figure 5B,G).

### 2.7. AET Improves the Functional Capacity and Metabolic Profile of Exercise in AoS Animals

Figure 6A–G presents the assessment of cardiorespiratory fitness. AoS caused a decrease in exercise tolerance compared to the respective Sham in the four stress tests, as verified by the decline in the exhaustion velocity and LT velocity variables (Figure 6A–C), except in the initial T1 and T3 in the second variable mentioned. In addition, the Sham-ET group showed greater cardiorespiratory fitness than the Sham at T2, T3, and T4 (Figure 6C).

The AoS group demonstrated a higher concentration of basal lactate than the Sham group. In comparison, the AoS-ET animals presented low levels of basal lactate compared to the AoS rats.

AoS animals showed an increase in the relative amount of lactate at LT and exhaustion compared to Sham animals (Figure 6F,G). The AoS-ET group showed changes in relation to Sham-ET only in the relative lactate concentration at the lactate threshold in T2. At T2, T3, and T4, there was a reduction in the relative amount of lactate at the lactate threshold and exhaustion in the AoS-ET group compared to the AoS (Figure 6F,G). These outcomes show that AoS reduced the aerobic capacity of the animals since the production of lactate, a marker of the activation of anaerobic metabolism, was increased at the mentioned moments compared to the control animals. On the other hand, the AET efficiently increased the aerobic capacity of the animals with heart disease, producing less lactate at the lactate threshold and exhaustion when corrected by the velocities at the respective moments.

## 3. Discussion

Our study aimed to evaluate the effects of AET on Ca^2+^ handling and cardiac function in rats with heart failure due to AoS. The findings showed that animals with AoS developed heart failure; also, AET generated positive adaptations in physical fitness, the heart, and mechanisms of Ca^2+^ sarcoplasmic reuptake and cellular influx in the myocardium in rats. In these animals, AET was able to: (1) improve aerobic potential and functional capacity; (2) enhance left ventricular systolic and diastolic function and decrease the frequency of signs of heart failure; (3) decrease tension during myocardial rest and attenuate the loss of calcium reuptake potential by the sarcoplasmic reticulum; (4) improve the functional capacity of cardiomyocytes, favoring positive adjustments in cardiomyocyte contractility and relaxation; and (5) increase the sensitivity of myofilaments to Ca^2+^.

The magnitude of systolic and diastolic cardiac dysfunction and the structural changes verified in the echocardiographic exams after 18 and 28 weeks of the experimental protocol are in agreement with previous studies [11,12,27,28,29,30,31,32,33]. The structural adequacy, evidenced by cardiac hypertrophy after a period of aortic constriction, is not functional, presenting a pathological characteristic. Several pathophysiological mechanisms contribute to the dysfunctional hypertrophy response, such as disorganization of the extracellular matrix; decreased adrenergic responsiveness; changes in contractile proteins, cytoskeleton, and energy metabolism; loss of myocytes by necrosis, apoptosis, or autophagy; reprogramming; and impairments in the excitation/contraction/relaxation coupling process [34,35,36,37,38,39,40,41,42,43]. Furthermore, the depression of cardiac function was accompanied by the appearance of clinical and pathological signs of heart failure, including a significant loss in body weight, exercise intolerance, features of cachexia [30], increased right ventricular water content, altered breathing pattern, ascites, pleural effusion, atrial thrombus, and hepatic congestion. Findings from in vitro analyses support the outcomes of the 28-week echocardiographic examination. The animals submitted to AoS had myocardial stiffness (↑RT) and impaired ability to develop tension (↓+dT/dt and DT), shorten (↓Fractional shortening, SMV and ↑TS_50%_), and relax (↑TR_50%_). Previous studies are in agreement with the data obtained in our research [12,14,44].

Due to the complexity of the pathophysiological process, the treatment of heart failure has been one of the great challenges of the scientific community in recent decades, and therapeutic success and/or positive prognosis depend on the type, time, and intensity of overload and individual characteristics [38,45,46]. In addition to the drugs tested in different research studies and used in the treatment of heart failure, physical activity presents a relevant potential to attenuate and/or reverse the transition from pathological hypertrophy to heart failure [35,36,47]. In this sense, AET has been consolidated as a therapeutic tool in managing cardiovascular diseases, both in the preventive context and as a complementary treatment of heart diseases [1,6,8,9,10,16,35,48,49].

Similar to previous studies, low-volume and moderate-intensity AET were implemented at 18 weeks of AoS after the establishment of ventricular dysfunction in the animals. The low exercise tolerance of animals with severe pressure overload justifies the protocol type [11,12,29,30,31,32,33]. This fact was confirmed in the first cardiorespiratory test after 18 weeks of surgery, in which the animals with heart disease showed exercise intolerance, as visualized by the reduced exhaustion velocity in relation to the respective Sham groups. During and at the end of the experimental protocol, the AoS-ET animals showed significant improvement in exercise tolerance and lactate levels at different intensities of effort, demonstrating the exercise efficiency of the AoS group. The rise in cardiac and musculoskeletal aerobic potential by AET was evidenced by the observation of lower relative lactate concentrations at the moments of lactate threshold and exhaustion [30,33,50,51]. In addition, the higher speed of exhaustion of the exercised groups expresses enhanced functional capacity and quality of life in the animals.

In this study, AET attenuated the systolic dysfunction and, mainly, the diastolic damage resulting from the pressure overload imposed by the AoS, as in previous studies [11,12,29,30,31,32,33]. Trained rats with heart disease showed higher values of EF, PWST, EDT, E′, and A′ and lower values of LA/Ao, E wave, and E/A and E′/A′ ratios compared to the sedentary group; these outcomes point to an increase in the contractile capacity and less myocardial stiffness in the AoS animals. In the literature, the AET prevented heart failure in models of myocardial infarction in rats [17,22,23,25], genetic sympathetic hyperactivity in mice [15,24], and heart rate overload by the ventricular pacemaker in dogs [19]; authors related the benefits to the improvement of Ca^2+^ handling and its regulatory agents. Other studies using mice with AoS found worsening of ventricular dysfunction associated with changes in Ca^2+^ transit regulatory proteins [26], increased collagen deposition [26,52], and oxidative stress [52] after eight weeks of voluntary exercise, starting immediately after surgery. These authors proposed that aortic stenosis blocks the beneficial vasodilatory effect of AET via activation of endothelial nitric oxide synthase (eNOS) due to fixed aortic obstruction [52]. However, in our study, even without modifying the aggressor agent (aortic stenosis), important systolic and diastolic benefits were detected in the heart function, making it possible to infer that the positive adaptations result in intrinsic improvement of cells and cardiac tissue constituents. The in vitro study is in line with the echocardiogram results. AET decreased myocardial stiffness (↓RT) and increased the ability to shorten (↑Fractional shortening and SMV) and relax (↓TR_50%_). An important study with infarcted rats showed amelioration of fractional shortening and relaxation time in cardiomyocytes from trained animals compared to sedentary animals [25].

Studies have shown, in different experimental models, that Ca^2+^ handling adjustments are essential for enhancing the heart’s performance by the AET in heart failure and dysfunction [14,15,16,17,18,19,20,21,22,23,24,25]. According to Kim et al. [45], the restoration of protein expression levels related to the excitation–contraction–relaxation coupling close to the normal heart is among the positive adaptations to the therapies implemented during heart failure. Our results confirm the proposition since AoS-ET animals showed lower protein expression of SERCA2a and NCX compared to AoS, and these values were similar to the control groups; however, the data are in disagreement with the literature, which mostly points to maintenance or increase in SERCA2a expression by AET in normal, infarcted, and hypertensive animals [16,18,25,53]. In pigs with AoS and preserved ejection fraction submitted to the exercise, authors observed a raised SERCA2a/PLB ratio, serine 16 phosphorylated PLB, and NCX, in addition to decreased diastolic Ca^2+^, which were associated with functional improvements of cardiomyocytes compared to sedentary animals [14]. However, van Deel et al. [26] did not identify improvement in cardiac function and positive adjustments in Ca^2+^ handling by voluntary training in mice with AoS, possibly due to the model and severity of AoS and the exercise protocol. It is noteworthy that, in our study, the reduced SERCA2a in trained AoS animals was accompanied by an improvement in cardiomyocyte function in relation to AoS animals; moreover, these rats expressed beneficial values for RMV and TR50% compared to the sedentary animals in the analysis of isolated cardiomyocytes. In addition, the AoS-ET group improved SMV and shortening fraction, which may be related to the higher sensitivity of myofilaments to Ca^2+^, as proposed by the improvement in the ratios of systolic Ca^2+^/Fractional shortening and systolic Ca^2+^/SMV. In agreement, previous studies have shown increased sensitivity of myofilaments to Ca^2+^ by exercise training in normal [54,55] and infarcted [25] rats.

The maneuvers performed during the isolated papillary muscle analysis provided support regarding the effects of AET on SERCA2a and L-channel activity in AoS animals. In the blockage of the ATP binding site of SERCA2a by cyclopiazonic acid, there was no statistical difference between the AoS-ET and AoS groups for all variables and moments studied. As there was a significant decrease in SERCA2a expression in the sarcoplasmic reticulum from AoS-ET mice in relation to the AoS group, the similarity of response suggests an enhanced function of SERCA2a in the AoS-ET group compared to the AoS group; this group should have presented better performance after the electrical stimulus pauses and a lower percentage of inhibition after ACP, due to the increased amount of SERCA2a. Authors have shown augmented SERCA2a activity promoted by interval aerobic exercise in normal [53,56] and infarcted [25] rats. As previously mentioned, this protein is an ATPase, which depends on cytosolic ATP levels to perform its normal function [57]; thus, the number and functioning of mitochondria play a fundamental regulatory role [58]. The literature shows that AET improves mitochondrial function, which is impaired in heart diseases [59], as well as decreases oxidative stress [60], which is raised in pathological conditions [38]. Therefore, in the present study, these mechanisms may have been related to the attenuation of SERCA2a functional impairments.

We believe that AET also generates adjustments in the Ca^2+^ entry mechanism in diseased cardiomyocytes since, in the diltiazem blockade experiment, there was a better performance of AoS-ET animals compared to AoS rats for the variables DT, +dT/dt, and –dT/dt. The literature differs regarding the effects of exercise training on Ca^2+^ current (I_Ca_), which is governed by L-channels in normal rats [53,56,61,62,63]. While authors have shown increased protein expression [56] and adaptive plasticity of L-channels [63], other studies have not detected a training effect on I_Ca_ or this protein [61,62].

## 4. Materials and Methods

### 4.1. Study Design

Twenty one-day-old male Wistar rats were submitted to simulated (Sham, *n* = 61) or aortic stenosis induction (AoS, *n* = 44) surgery. After 18 weeks of experimental protocol, the animals were divided into four groups regarding the practice or not of AET for a period of 10 weeks: simulated surgery (Sham, *n* = 28); Sham plus exercise training (Sham-ET, *n* = 33); AoS surgery (AoS, *n* = 22); and AoS plus exercise training (AoS-ET, *n* = 22).

The animals were subjected to two lines of experimental analysis:The first experimental line evaluated cardiac function by echocardiogram, isolated papillary muscles, and the expression of Ca^2+^-handling regulatory protein by Western Blot. SERCA2a and L-type Ca^2+^ channels activity were analyzed during post-rest contraction and Ca^2+^ elevation (Sham, *n* = 22; Sham-ET, *n* = 20; AoS, *n* = 16; AoS-ET, *n* = 16), respectively, and by the cumulative administration of extracellular Ca^2+^ in the presence of SERCA2a or L-type Ca^2+^ channels specific blockers in the isolated papillary muscle assay. In addition, to assess functional capacity and prescribe AET, the animals were submitted to cardiorespiratory fitness tests at weeks 18, 22, 25, and 28 of the experimental protocol.The second experimental line evaluated cardiac function by echocardiogram and isolated cardiomyocyte assay, which measured mechanical function and Ca^2+^ handling (Sham, *n* = 6; Sham-ET, n = 6; AoS, *n* = 6; AoS-ET, *n* = 6). The week 28 cardiorespiratory fitness tests were not performed on these animals due to technical issues.

As noted, the echocardiogram evaluation contemplated all animals in the study (Sham, *n* = 28; Sham-ET, *n* = 33; AoS, *n* = 22; AoS-ET, *n* = 22).

### 4.2. Animals

The Wistar rats obtained from the Animal Center of Botucatu Medical School (Botucatu, São Paulo, Brazil) were allocated in collective cages at a 23 °C room temperature, with a 12 h light/dark cycle, relative humidity of 60%, and water ad libitum. The research was approved by the “Committee for Experimental Research Ethics of the Faculty of Medicine in Botucatu—UNESP”, in accordance with the “Guide for the Care and Use of Laboratory Animals” (protocol 1138/2015).

### 4.3. AoS Surgery

AS was surgically induced, as previously described [64,65]. Briefly, rats were anesthetized with a mixture of ketamine hydrochloride (50 mg/kg i.p.) and xylazine hydrochloride (10 mg/kg, i.p.), and the heart was exposed through a median thoracotomy. Then a stainless-steel clip (0.60 mm internal diameter) was placed in the ascending aorta, approximately 3 mm from its root. During surgery, the rats received 1 mL of warm saline intraperitoneally and were manually ventilated with positive pressure and 100% oxygen. After the procedure, rats were kept warm until full consciousness was regained. The analgesia procedure consisted of intraperitoneal administration of carprofen (5mg/kg body weight) and was maintained until the disappearance of evidence of pain. Control animals were subjected to the same procedure but without the constriction of the aorta.

Rats were anesthetized with a mixture of ketamine hydrochloride (60 mg/kg i.p.) and xylazine hydrochloride (10 mg/kg, i.p.) and then euthanized by decapitation. Heart tissue was dissected, and then the left ventricle (LV) was removed and immediately freeze-clamped at the temperature of liquid nitrogen. Blood samples were collected, and the serum was separated by centrifugation at 1620× *g* for 10 min at 4 °C.

### 4.4. Cardiorespiratory Fitness Test (CFT)

The prescription of AET and the functional capacity assessment were performed using the cardiorespiratory fitness test (CFT). After one week of adaptation to the treadmill, the Sham, Sham-ET, AoS, and AoS-ET groups underwent CFT. Before the beginning of the AET, after 18 weeks of surgery, the CFT was performed (T1) to prescribe the initial training workload. After the 4th and 7th weeks of AET (T2 and T3), CFTs were performed to readjust the training workloads. At the end of the 10th week of AET, CFT was performed again to assess the final physical fitness level of the animals (T4).

The exercise test started at a speed of 6 m/min, progressively increasing (3 m/min) after 3 min until exhaustion, as described previously [66]. The exhaustion of the animal defined the end of the test; the criterion adopted was the non-maintenance of the race at the imposed speed for 5 s. During the tests, 25 μL of blood was collected from the animal’s tail to analyze lactate concentrations at baseline and after each speed increment. For the collection, glass slides, 25 × 75 × 1 mm (Sigma Chemical Company^®^ USA, model Techware S8902, St. Louis, MO, USA), micropipettes, 20 to 200 μL, (Nichiryo Co.^®^ Japão, Model Nichipet NPX 200, Tokyo, Japan), and disposable tips were used. Blood samples were stored in 0.5 mL Eppendorf tubes containing 50 μL of 1% sodium fluoride and kept in a freezer until the analysis period. Lactate concentration was determined by the electro-enzymatic method using a lactimeter (Yellow Springs Instruments^®^, 2300 Stat Plus Glucose & L-Lactate Analyzer, Yellow Spring, OH, USA).

The AET prescription was based on the analysis of lactate curves, with the daily training speed equivalent to the lactate threshold (LT). This point was determined through the graphic plotting of lactate concentrations versus the running speed of the stages. The moment when there was a change in the linearity of the curve as a function of the increase in speed, established by visual inspection, was considered the LT [33]. The variables analyzed to determine the functional capacity of the animals were: velocities of exhaustion at the lactate threshold (LT velocity); lactate concentrations at baseline ([LAC]_basal_) and lactate threshold ([LAC]_LT_); and the ratios of lactate concentrations at lactate threshold and at exhaustion by respective velocities at each moment ([LAC]_(LT)_/Vel_(LT),_ and [LAC]_(Ex)_/Vel_(Ex)_).

### 4.5. Aerobic Exercise Training (AET)

AET was started 18 weeks after surgery and involved a rat-specific treadmill running program (Insight Instrumentos—Ribeirão Preto, São Paulo, Brazil) five days a week for 10 consecutive weeks [29,32,33]. The velocity equivalent to the LT was determined as the training intensity. The daily volume progressively increased during AET, presenting the following characteristics: 10, 12, 14, and 16 min in duration in weeks 1, 2, 3, and 4, respectively, and 16, 18, and 20 min in weeks 5, 6, and 7, respectively. AET volume was maintained at 20 min until the end of 10 weeks.

### 4.6. Cardiac Function

#### 4.6.1. Echocardiogram

The echocardiographic study provided data on cardiac structure and function before and after AET, i.e., after 18 and 28 weeks of experimental surgery. Commercially available echocardiography (General Electric Medical Systems, Vivid S6, Tirat Carmel, Israel) equipped with a 5–11.5 MHz multifrequency probe was used as previously described [29,31,64]. Rats were anesthetized via intraperitoneal injection with a mixture of ketamine (50 mg/kg; Dopalen^®^, Sespo Indústria e Comércio Ltd.a- Divisão Vetbrands, Jacareí, São Paulo, Brazil) and xylazine (0.5 mg/kg; Anasedan^®^, Sespo Indústria e Comércio Ltd.a- Divisão Vetbrands, Jacareí, São Paulo, Brazil). The following variables evaluated cardiac structure: LA normalized to the aortic diameter (LA/Ao), left ventricle diastolic diameter (LVDD), left ventricular systolic diameter (LVSD), posterior wall diastolic thickness (PWDT), interventricular septum diastolic thickness (ISDT), and relative wall thickness (RWT). The following parameters assessed ventricular function: heart rate (HR), midwall fraction shortening (FS), ejection fraction (EF), posterior wall systolic velocity (PWSV), early diastolic mitral inflow velocity (E wave), ratio between E wave and atrial contraction flow peak (A wave), E-wave deceleration time (EDT), velocity of the mitral annulus during early ventricular filling (E′), mitral velocity annulus during atrial contraction (A′), and the ratio between filling flow peak and mitral annulus velocity during early ventricular filling (E/E′).

#### 4.6.2. Isolated Papillary Muscle Assay

The cardiac contractile performance was evaluated by studying isolated papillary muscles from the LV as previously described [27,64]. The following mechanical parameters were measured during isometric contraction: maximum developed tension (DT; g/mm^2^), maximum rate of tension development (+dT/dt; g/mm^2^/s), and decline (−dT/dt; g/mm^2^/s). Regulator mechanisms of Ca^2+^ influx and L-type calcium channel activity were analyzed by Ca^2+^ concentration extracellular elevation maneuver (response percentage) and elevation of extracellular Ca^2+^ concentrations (0.5, 1.5, 2.5, and 3.5 mM) in the presence and absence of diltiazem (10^−5^ M; Diltiazem Hydrochloride, Sigma^®^ Aldrich, St Louis, MO, USA), a specific blocker of L-type calcium channels. A post-rest contraction maneuver (response percentage) and elevation of extracellular Ca^2+^ concentrations (0.5, 1.5, 2.5, and 3.5 mM) in the presence and absence of cyclopiazonic acid (CPA, 30 mM; Penicillium cyclopium, Sigma^®^ Aldrich, St Louis, MO, USA), a highly specific blocker of SERCA2a, were performed to assess the potential of SERCA2a function. All variables were normalized per cross-sectional area of the papillary muscle (CSA). Seven papillary muscles with CSA >1.5 mm^2^ were excluded from analysis as they can present central core hypoxia and impaired functional performance [64,65].

#### 4.6.3. Isolated Cardiomyocyte Assay

Cardiomyocyte Contractility

Under anesthesia, rats from each group were euthanized, and the hearts were quickly removed by median thoracotomy and enzymatically isolated as previously described [27]. Briefly, the hearts were cannulated, and retrograde perfusion of the aorta was performed in the Langendorff system (37 °C) with a modified isolation digestion buffer solution (DB), a calcium-free solution containing 0.1 mM ethylene glycol-bis (ß-aminoethyl ether)-N, N, N′, N′- tetraacetic acid (EGTA), and N-[2-hydro-ethyl]-piperazine-N′-[2-ethanesulfonic acid] (HEPES) equilibrated with 5% CO_2_ and 95% O_2_ for ~3 to 5 min. The composition of the DB solution was (mM): 130 NaCl, 1.4 MgCl2, 5.4 KCl, 25 HEPES, 22 glucose, 0.33 NAH2PO4, and pH 7.39. Afterwards, the hearts were perfused for 20–30 min with a DB solution containing 1 mg/mL collagenase type II (Worthington Biochemical Corporation, Lakewood, NJ, USA) and Ca^2+^ (1 mM). Subsequently, isolated cells were placed in an experimental chamber with a glass coverslip base mounted on the stage of an inverted microscope (MyoCam-S, IonOptix, Milton, MA, USA) with an edge detection system with a 40× objective lens (Nikon Eclipse—TS100, USA). After the digestion process, the supernatant was removed, and the myocytes were resuspended in Tyrode’s buffer containing (in mM): 140 NaCl, 10 HEPES, 0.33 NaH2PO4, 1 MgCl2, 5 KCl, 1.8 CaCl2, and 10 glucose. Cells were immersed in Tyrode’s solution and field stimulated at 1 Hz (20 V, 5 ms duration square pulses). Cell shortening in response to electrical stimulation was measured with a video-edge detection system at a 240 Hz frame rate (Ionwizard, Ion Optix, Milton, MA, USA), and the contractile parameters were evaluated. Sarcomere length, fractional shortening (expressed as a percentage of resting cell length), maximum shortening velocity, maximum relaxation velocity, and time to 50% shortening (time to 50% peak) and 50% relaxation (time for 50% relaxation) were measured in 6 cells per animal in each experimental group.

Intracellular Ca^2+^ Measurements

Myocytes were loaded with 1.0 μM Fura2-acetoxymethyl (AM) ester (Molecular Probes, Eugene, OR, USA) for 10 min at room temperature, washed with Tyrode solution, and allowed to rest for an additional 10 min to allow the de-esterification of dye. Subsequently, the cardiomyocytes were stimulated at 1 Hz, and fluorescence images were obtained using excitation of 340 to 380 nm wavelengths using a Hyper Switch system (Ionwizard, IonOptix, Milton, MA, USA). Background-corrected Fura 2 AM ratios, reflecting intracellular Ca^2+^ concentration detected at approximately 510 nm. Diastolic and systolic Ca^2+^, time to Ca^2+^ peak, and time to 50% Ca^2+^ peak and decay were also analyzed. In addition, the ratios of systolic Ca^2+^ by shortening fraction and by shortening maximum velocity were performed to evaluate the responsiveness of myofilaments to Ca^2+^.

### 4.7. Expression of Ca^2+^ Handling Protein

The protein expression of the elements responsible for the regulation of Ca^2+^ handling was analyzed by Western Blot. Fragments of the LV were frozen in liquid nitrogen and stored at −80 °C. Frozen samples were subsequently homogenized in RIPA buffer containing protease (SigmaAldrich, St. Louis, MO, USA) and phosphatase (Roche Diagnostics, Indianapolis, IN, USA) inhibitors using a bead beater homogenizer (Bullet Blender^®^, Next Advance, Inc., Troy, NY, USA). The homogenized product was centrifuged (5804R Eppendorf, Hamburg, Germany) at 12,000× *g* rpm for 20 min at 4 °C, and the supernatant was transferred to Eppendorf tubes and stored at −80 °C. Protein concentration was determined using the Pierce BCA Protein Assay Kit. SDS-PAGE was used to resolve 25 µg of protein lysate from each sample. Electrophoresis was performed with biphasic gel stacking (240 mm Tris-HCl pH 6.8, 30% polyacrylamide, APS, and TEMED) and resolving (240 mm Tris-HCl pH 8.8, 30% polyacrylamide, APS, and TEMED) at a concentration of 6 to 10%, depending on the molecular weight of the analyzed protein. The Kaleidoscope Prestained Standard (Bio-Rad, Hercules, CA, USA) was used to identify band sizes. Electrophoresis was performed at 120 V (Power Pac HC 3.0 A, Bio-Rad, Hercules, CA, USA) for 3 h with running buffer (0.25 M Tris, 192 mM glycine, and 1% SDS). Proteins were transferred to a nitrocellulose membrane (Armsham Biosciences, Piscataway, NJ, USA) using a Mini Trans-Blot (Bio-Rad, Hercules, CA, USA) system with transfer buffer (25 mM Tris, 192 mM glycine, 20% methanol, and 0.1% SDS). Membranes were blocked with 5% non-fat dry milk in TBS-T buffer (20 mM Tris-HCl pH 7.4, 137 mM NaCl, and 0.1% Tween 20) for 120 min at room temperature under constant agitation. The membrane was washed three times with TBS-T and incubated for 12 h at 4–8 °C under constant agitation with the following primary antibodies: SERCA2a (1:2500; ABR, Affinity BioReagents, Golden, CO, USA), Phospholamban (1:5000; ABR), Phospho-Phospholamban (Ser16) (1:5000; Badrilla, Leeds, West Yorkshire, UK), Phospho–Phospholamban (Thr17) (1:5000; Badrilla), NCX (1:2000; Upstate, Lake Placid, NY, USA), Calcium Channel, VoltageGated Alpha 1C (1:100; Chemicon International, Temecula, CA, USA), Ryanodine Receptor (1:5000; ABR, Affinity Bioreagents, Golden, CO, USA), and GAPDH (1:1000; Santa Cruz Biotechonology Inc., Santa Cruz, CA, USA). After incubation with the primary antibody, membranes were washed three times in TBS-T and incubated with peroxidase-conjugated secondary antibodies (anti-rabbit or anti-mouse IgG; 1:5000–1:10,000; Abcam, Waltham, MA, USA) for 2 h under constant agitation. Membranes were then washed three times with TBS-T to remove excess secondary antibodies. Blots were incubated with ECL (Enhanced Chemi-Luminescence, Amersham Biosciences, Piscataway, NJ, USA) for chemiluminescence detection by ImageQuant™ LAS 4000 (GE Healthcare, Salt Lake City, UT, USA). Quantification analysis of blots was performed using Scion Image software (Scion Corporation, Frederick, MD, USA). The immunoblots were quantified by densitometry using ImageJ Analysis software 1.53t (NIH), and target band results were normalized to the expression of cardiac GAPDH [12]. The RyR is expressed without normalization because it is not possible to evaluate GAPDH as a normalizer in the same gel as the RyR due to the difference in molecular weight between the two proteins.

### 4.8. Statistical Analysis

The Kolmogorov–Smirnov test was used to evaluate the data’s normality. Data from cardiorespiratory fitness tests, cardiac function, in vivo and in vitro analysis, and Ca^2+^ handling protein expression are reported as means ± standard deviation (SD), or median (25 percentile; 75 percentile). Comparisons among four groups were evaluated using two-way analysis of variance (ANOVA) for independent samples or Kruskal–Wallis, complemented with Bonferroni or Dunn’s multiple comparison tests, respectively. Ca^2+^ elevation, post-rest contraction, and SERCA2a and L-channel blockage maneuvers are reported as means ± SD and studied using analysis of variance for repeated measures, complemented with Bonferroni’s multiple comparison test. The data were evaluated at a significance level of 0.05. The statistical analyses were performed using SigmaStat 3.5, and graphics were generated using GraphPad Prism 5.

## 5. Conclusions

AET improves myocardial Ca^2+^ handling in animals with supravalvular AoS and ventricular dysfunction since in vitro cardiac performance analysis showed improvement in the capacity of Ca^2+^ recapture by the sarcoplasmic reticulum, consequent to the positive adaptations of the SERCA2a and L-type Ca^2+^ channel activities, and increased responsiveness of myofilaments to Ca^2+^, which supports the positive responses observed in echocardiograms and cardiorespiratory fitness tests. Our findings reinforce the importance of controlled exercise training not only in health promotion and disease prevention but also as a possible tool in treating heart failure. The loss of quality of life and functional capacity of individuals with heart failure is evident, and it is essential to develop strategies to improve this scenario and ensure greater longevity for this population. In addition, full knowledge of the molecular mechanisms related to the benefits of different types and intensities of exercise in failing hearts is fundamental for managing the intervention, which must be specific for each type and severity of cardiac pathology. Cardiac improvement from a macro point of view—the heart as a pump—is a product of the positive adjustments observed on a micro level—a molecular and cellular view. An enhanced heart function, along with the improvement of other organs by exercise training, generates a higher quality of life for individuals with heart failure.

## Figures and Tables

**Figure 1 ijms-24-12306-f001:**
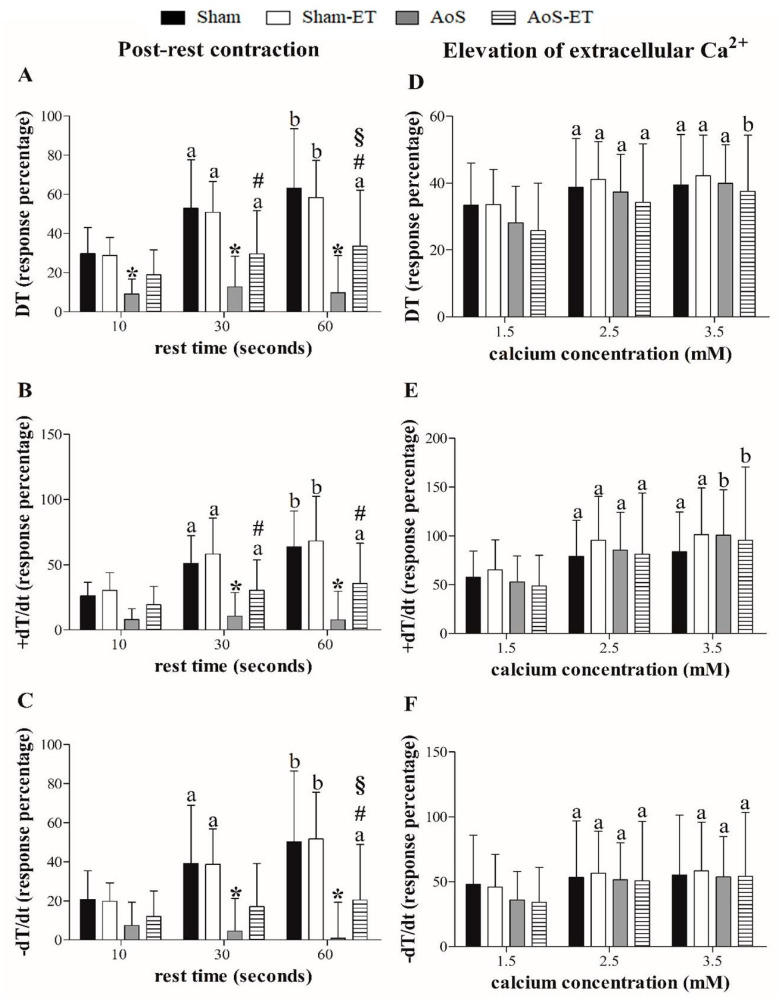
Response percentage to post-rest contraction (**A**–**C**) and elevation of extracellular calcium (Ca^2+^) concentration (**D**–**F**) from baseline (Ca^2+^ concentration: 0.5 mM). DT: maximum developed tension; +dT/dt: maximum rate of tension development; −dT/dt: maximum rate of tension decline. Data are expressed as means ± SD of maneuver response percentage. Sham: animals submitted to simulated surgery (*n* = 22); Sham-ET: animals submitted to simulated surgery and aerobic exercise training (AET) (*n* = 20); AoS: animals submitted to aortic stenosis (AoS) surgery (*n* = 16); AoS-ET: animals submitted to AoS surgery and AET (*n* = 16). Analysis of variance (ANOVA) for repeated measures and Bonferroni post hoc test. *p* < 0.05. ^a^ vs. 10 s, ^b^ vs. 10 s and 30 s, ^a^ vs. 1.5 Ca^2+^, ^b^ vs. 1.5 and 2.5 Ca^2+^, * AoS vs. Sham, ^#^ AoS-ET vs. Sham-ET, ^§^ AoS-ET vs. AoS.

**Figure 2 ijms-24-12306-f002:**
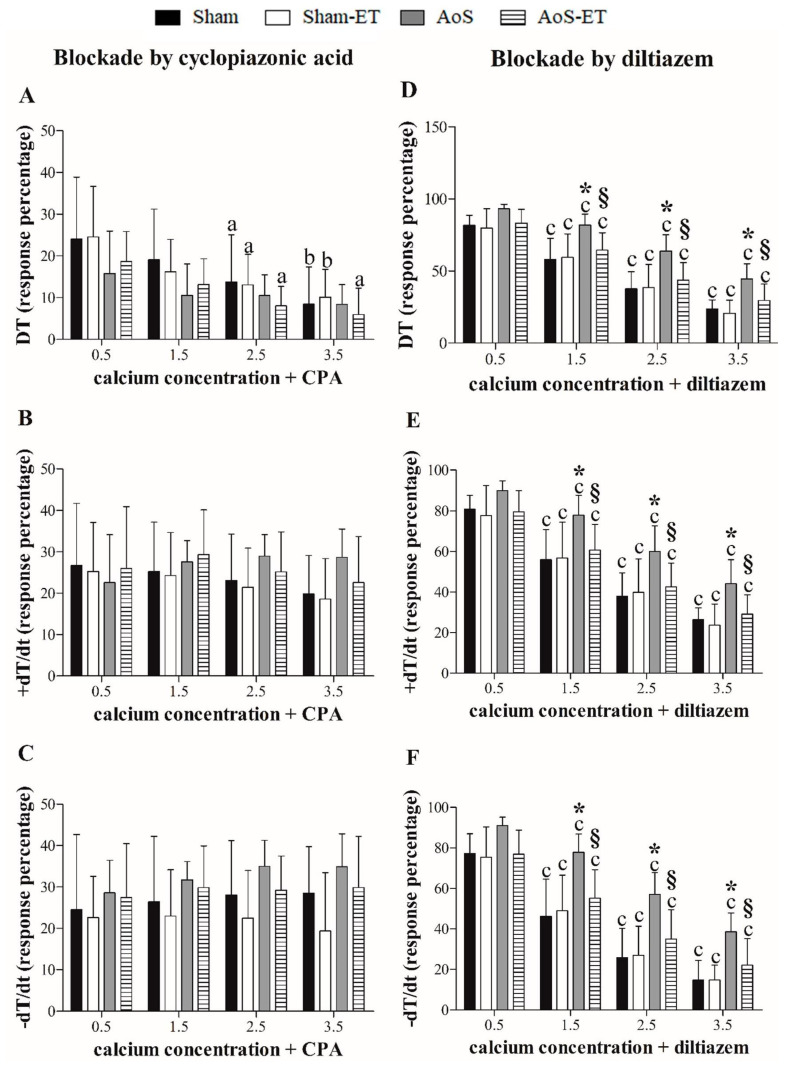
Inhibition percentage of DT (maximum developed tension), +dT/dt (maximum rate of tension development), and −dT/dt (maximum rate of tension decline) to cyclopiazonic acid (SERCA2a blocker; (**A**–**C**) and diltiazem (L-type calcium channels blocker; (**D**–**F**) plus incremental calcium concentration. Data are expressed as means ± SD. Sham: animals submitted to simulated surgery (*n* = 11); Sham-ET: animals submitted to simulated surgery and aerobic exercise training (AET) (*n* = 10); AoS: animals submitted to aortic stenosis (AoS) surgery (*n* = 8); AoS-ET: animals submitted to AoS surgery and AET (*n* = 8). Analysis of variance (ANOVA) for repeated measures and Bonferroni post hoc test. *p* < 0.05. ^a^ vs. 0.5 Ca^2+^, ^b^ vs. 0.5 and 1.5 Ca^2+^, ^c^ vs. 0.5, 1.5, and 2.5 Ca^2+^, * AoS vs. Sham, ^§^ AoS-ET vs. AoS.

**Figure 3 ijms-24-12306-f003:**
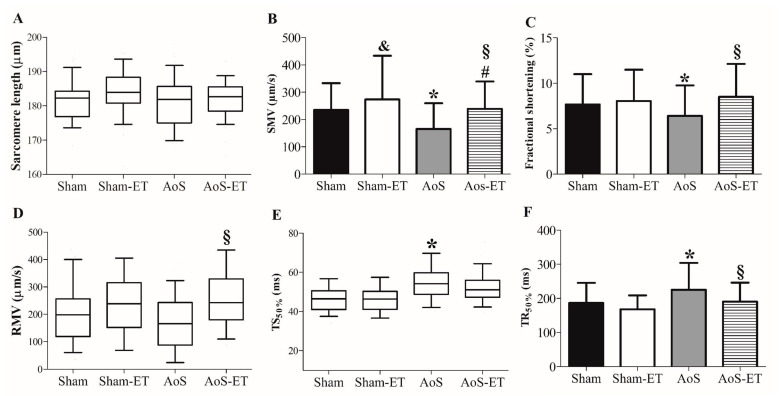
Cardiomyocyte mechanical function (**A**–**F**). SMV: shortening maximum velocity; RMV: relaxation maximum velocity; TS_50%_: time to 50% shortening; TR_50%_: time to 50% relaxation. Data are expressed as means ± SD or median (25 percentile; 75 percentile). Sham: animals submitted to simulated surgery (*n* = 6; number of cells = 71); Sham-ET: animals submitted to simulated surgery and aerobic exercise training (AET) (*n* = 6; number of cells = 76); AoS: animals submitted to AoS surgery (*n* = 6; number of cells = 96); AoS-ET: animals submitted to AoS surgery and AET (*n* = 6; number of cells = 100). Analysis of variance (ANOVA) and Bonferroni post hoc test or Kruskal–Wallis and Dunn’s method post hoc. *p* < 0.05. ^&^ Sham vs. Sham-ET; * AoS vs. Sham; ^#^ AoS-ET vs. Sham-ET; ^§^ AoS-ET vs. AoS.

**Figure 4 ijms-24-12306-f004:**
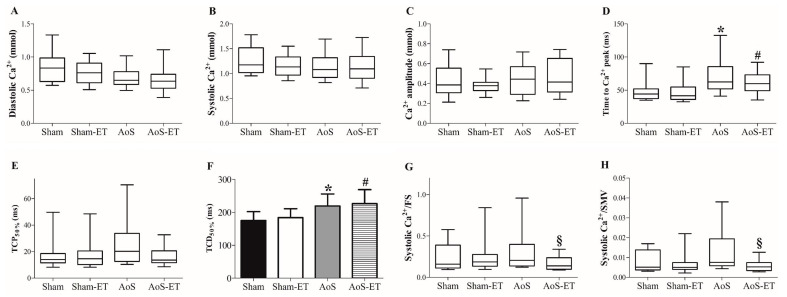
Cardiomyocyte calcium handling (**A**–**H**). TCP_50%_: time to 50% of Ca^2+^ peak; TCD_50%_: time to 50% of Ca^2+^ decay; SMV: shortening maximum velocity; RMV: relaxation maximum velocity. Data are expressed as means ± SD or median (25 percentile; 75 percentile). Sham: animals submitted to simulated surgery (*n* = 6; number of cells = 29); Sham-ET: animals submitted to simulated surgery and aerobic exercise training (AET) (*n* = 6; number of cells = 36); AoS: animals submitted to AoS surgery (*n* = 6; number of cells = 41); AoS-ET: animals submitted to AoS surgery and AET (*n* = 6; number of cells = 46). Analysis of variance (ANOVA) and Bonferroni post hoc test or Kruskal–Wallis and Dunn’s method post hoc. *p* < 0.05. * AoS vs. Sham; ^#^ AoS-ET vs. Sham-ET; ^§^ AoS-ET vs. AoS.

**Figure 5 ijms-24-12306-f005:**
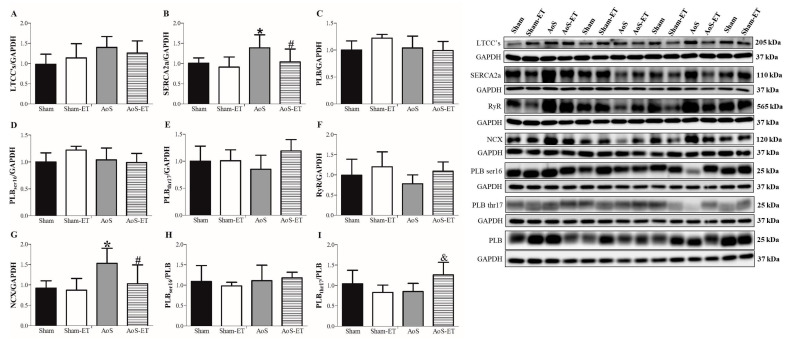
Expression of calcium handling protein (**A**–**I**). RyR: ryanodine receptor; SERCA2a: Sarcoplasmic Reticulum Calcium Pump; NCX: Sodium and calcium exchanger; PLB: Phosfolamban; PLBser16: Phosfolamban phosphorylated at serine 16; PLBthr17: Phosfolamban phosphorylated on threonine 17; CSQ: Calsequestrin; LTCCs: L-type calcium channels. Data are expressed as means ± SD. Sham: animals submitted to simulated surgery (*n* = 7); Sham-ET: animals submitted to simulated surgery and aerobic exercise training (AET) (*n* = 7); AoS: animals submitted to AoS surgery (*n* = 7); AoS-ET: animals submitted to AoS surgery and AET (*n* = 7). Analysis of variance (ANOVA) and Bonferroni post hoc test. *p* < 0.05. * AoS vs. Sham; ^&^ AoS-ET vs. AoS; ^#^ AoS-ET vs. Sham-ET.

**Figure 6 ijms-24-12306-f006:**
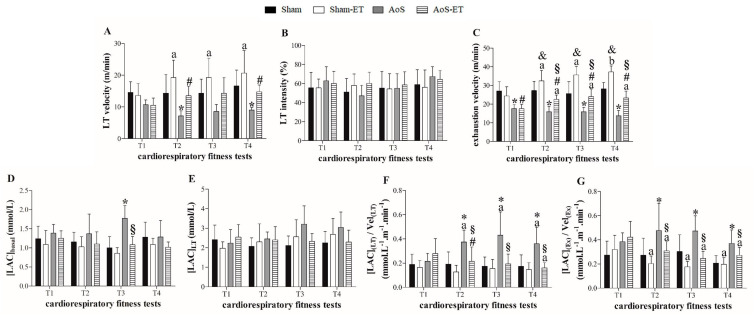
Cardiorespiratory fitness (**A**–**G**). T1: first cardiorespiratory fitness test; T2: second cardiorespiratory fitness test, after four weeks of aerobic exercise training (AET); T3: third cardiorespiratory fitness test, after seven weeks of AET; T4: fourth cardiorespiratory fitness test, after termination of the AET protocol. LT velocity: velocity at lactate threshold; LT intensity: intensity at lactate threshold; [LAC]_basal_: lactate concentration at baseline; [LAC]_LT_: lactate concentration at lactate threshold; [LAC]_(LT)_/Vel_(LT)_: ratio of lactate concentration at lactate threshold to lactate threshold velocity; [LAC]_(Ex)_/Vel_(Ex)_: ratio of lactate concentration at exhaustion-to-exhaustion velocity. Data are expressed as means ± SD. Sham: animals submitted to simulated surgery (*n* = 7); Sham-ET: animals submitted to simulated surgery and AET (*n* = 7); AoS: animals submitted to AoS surgery (*n* = 7); AoS-ET: animals submitted to AoS surgery and AET (*n* = 7). Analysis of variance (ANOVA) for repeated measures and Bonferroni post hoc test. *p* < 0.05. * AoS vs. Sham; ^&^ Sham vs. Sham-ET; ^§^ AoS-ET vs. AoS; ^#^ AoS-ET vs. Sham-ET. ^a^ vs. T1; ^b^ vs. T1 and T2.

**Table 1 ijms-24-12306-t001:** Echocardiogram data before and after aerobic exercise training (AET).

	Before AET				After AET			
	Sham	Sham-ET	AoS	AoS-ET	Sham	Sham-ET	AoS	AoS-ET
BW (g)	469 ± 38	472 ± 55	450 ± 54	452 ± 59	519 ± 45	494 ± 60	494 ± 70	487 ± 59
HR (bpm)	285 (280; 301)	290 (269; 301)	290 (262; 312)	290 (280; 301)	298 ± 42	292 ± 37	293 ± 43	305 ± 36
LVDD (mm)	7.41 (6.90; 7.66)	7.30 (6.90; 7.59)	7.88 (7.21; 8.46)	7.41 (6.32; 7.95)	7.41 (7.15; 7.66)	7.41 (7.09; 7.92)	8.43 (7.27; 9.13) **	7.92 (7.34; 8.49)
LVSD (mm)	3.07 (2.81; 3.32)	2.81 (2.55; 3.07)	3.58 (3.24; 4.02)	3.32 (2.74; 3.58)	3.07 (2.81; 3.32)	3.07 (2.81; 3.38)	3.65 (3.32; 5.40) **	3.58 (2.81; 4.52)
PWDT (mm)	1.53 (1.53; 1.53)	1.53 (1.53; 1.57)	3.07 (2.81; 3.07) *	2.92 (2.81; 3.07) ^#^	1.53 (1.53; 1.65)	1.53 (1.53; 1.61)	2.81 (2.55; 3.03) **	2.30 (2.55; 2.81) ^##^
ISDT (mm)	1.56 (1.53; 1.65)	1.55 (1.53; 1.65)	3.07 (2.98; 3.24) *	3.07 (3.07; 3.32) ^#^	1.62 (1.53; 1.70)	1.60 (1.53; 1.65)	2.95 (2.81; 3.07) **	2.81 (2.55; 3.07) ^##^
RWT	0.43 ± 0.03	0.43 ± 0.04	0.76 ± 0.13 *	0.81 ± 0.11 ^#^	0.43 ± 0.03	0.42 ± 0.03	0.68 ± 0.14 **	0.68 ± 0.11 ^##^
LA/Ao	1.21 (1.19; 1.30)	1.23 (1.19; 1.29)	1.93 (1.67; 2.00) *	1.87 (1.52; 2.11) ^#^	1.23 ± 0.09	1.24 ± 0.09	1.93 ± 0.19 **	1.79 ± 0.19 ^##§^
E wave (cm/s)	81 ± 7	83 ± 8	124 ± 24 *	127 ± 22 ^#^	85 ± 7	82 ± 9	130 ± 20 **	120 ± 25 ^##§^
E/A	1.64 ± 0.22	1.64 ± 0.25	3.55 ± 1.67 *	3.82 ± 2.00 ^#^	1.50 ± 0.20	1.54 ± 0.18	5.22 ± 1.42 **	3.82 ± 1.85 ^##§^
E′ (cm/s)	5.60 (5.30; 5.80)	5.60 (3.20; 6.10)	5.40 (3.70; 6.25)	5.40 (3.70; 6.12)	6.13 ± 0.77	6.20 ± 0.73	5.19 ± 0.99 **	5.40 ± 1.18 ^##^
A′ (cm/s)	3.75 (3.70; 3.90)	3.70 (3.35; 4.20)	3.30 (3.03; 5.35)	3.20 (3.20; 3.75)	4.24 ± 0.72	4.10 ± 0.60	3.08 ± 1.20 **	3.95 ± 1.50 ^§^
E/E′	14.3 ± 1.58	14.9 ± 1.98	25.9 ± 6.78 *	26.3 ± 7.82 ^#^	14.1 ± 2.24	13.4 ± 1.6	25.6 ± 4.85 **	22.5 ± 3.66 ^##§^
EDT (ms)	48 ± 7	48 ± 5	34 ± 11 *	36 ± 13 ^#^	49 ± 6	50 ± 4	28 ± 8 **	34 ± 11 ^##§^
PWSV (cm/s)	67 ± 9	71 ± 7	53 ± 12 *	55 ± 8 ^#^	68 ± 9	70 ± 9	37 ± 10 **	49 ± 9 ^##§^
FS (%)	26 ± 3	28 ± 4	25 ± 4	25 ± 4 ^#^	26 ± 3	25 ± 4	23 ± 5 **	24 ± 4
EF (%)	93 (91; 94)	94 (92; 95)	90 (87; 92)	89 (79; 92) ^#^	93 ± 2	92 ± 3	86 ± 9 **	90 ± 4 ^§^

Data are expressed as means ± SD or median (25 percentile; 75 percentile). Sham: animals submitted to simulated surgery (*n* = 28); Sham-ET: animals submitted to simulated surgery and aerobic exercise training (AET) (*n* = 33); AoS: animals submitted to aortic stenosis (AoS) surgery (*n* = 22); AoS-ET: animals submitted to AoS surgery and AET (*n* = 22). BW: body weight; HR: heart rate; LVDD: left ventricle (LV) diastolic diameter; LVSD: LV systolic diameter; PWDT: posterior wall diastolic thickness; ISDT: interventricular septum diastolic thickness; RWT: LV relative wall thickness; LA: left atrium; AO: aorta diameter; E/A: ratio between filling flow peak (E wave) and atrial contraction flow peak (A wave); EF: ejection fraction; FS: midwall fraction shortening; PWSV: posterior wall systolic velocity; E′: velocity of the mitral annulus during early ventricular filling; A′: mitral velocity annulus during atrial contraction; EDT: E-wave deceleration time; E/E′: ratio between filling flow peak and mitral annulus velocity during early ventricular filling. Analysis of variance (ANOVA) and Bonferroni post hoc test or Kruskal–Wallis and Dunn’s method post hoc. Before AET: * *p* < 0.05 vs. Sham, ^#^ *p* < 0.05 vs. Sham-ET. After AET: ** *p* < 0.05 vs. Sham, ^##^
*p* < 0.05 vs. Sham-ET, ^§^
*p* < 0.05 vs. AoS.

**Table 2 ijms-24-12306-t002:** Baseline data during the isolated papillary muscle study.

	Sham	Sham-ET	AoS	AoS-ET
CSA (mm^2^)	1.14 ± 0.13	1.15 ± 0.14	1.16 ± 0.16	1.20 ± 0.19
DT (g/mm^2^)	6.23 ± 1.43	6.12 ± 1.61	5.20 ± 1.11	5.23 ± 1.96
RT (g/mm^2^)	0.57 ± 0.19	0.61 ± 0.18	0.82 ± 0.22 *	0.66 ± 0.17 ^§^
+dT/dt (g/mm^2^/s)	66.6 ± 17.2	65.9 ± 18.1	46.9 ± 10.3 *	47.9 ± 20.0 ^#^
−dT/dt (g/mm^2^/s)	21.8 ± 5.50	22.1 ± 5.39	24.2 ± 5.57	23.0 ± 7.61
TPT (ms)	180 (180; 200)	180 (160; 180)	210 (182; 220)	200 (180; 210) ^#^

Data are expressed as means ± SD or median (25 percentile; 75 percentile). Sham: animals submitted to simulated surgery (*n* = 22); Sham-ET: animals submitted to simulated surgery and aerobic exercise training (AET) (*n* = 20); AoS: animals submitted to AoS surgery (*n* = 16); AoS-ET: animals submitted to AoS surgery and AET (*n* = 16). CSA: papillary cross-sectional area; DT: maximum developed tension; RT: resting tension; +dT/dt: maximum rate of tension development; -dT/dt: maximum rate of tension decline; TPT: time-to-peak tension. Analysis of variance (ANOVA) and Bonferroni post hoc test or Kruskal–Wallis and Dunn’s method post hoc. * *p* < 0.05 AoS vs. Sham; ^#^ *p* < 0.05 AoS-ET vs. Sham-ET; ^§^ *p* < 0.05 AoS-ET vs. AoS.

## Data Availability

All data were included in this manuscript. Additional datasets generated during and/or analyzed during the current study are available from the corresponding author upon request.

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
