# Peer review of "Aerobic Exercise Training Improves Calcium Handling and Cardiac Function in Rats with Heart Failure Resulting from Aortic Stenosis"

_ijms, 2023, doi:10.3390/ijms241512306_

Round 1
Reviewer 1 Report
This well-writing manuscript demonstrates the importance of aerobic excise training to reverse heart failure. The authors were trying to show the benefit of aerobic excise training on cardiac functional improvement and Ca2+ handling from animals, isolated papillary muscle, and isolated cardiomyocyte views. However, there are still some issues that need to be concerned.
1. The innovation of this manuscript is poor. The benefit of aerobic excise training on heart failure treatment and cardiac ca2+ handling has been widely reported in the past 20 years. Indeed, most of those studies used MI or other types of cardiac hypertrophy models, not for aortic stenosis, but the innovation of this study still needs to be improved. Authors have to figure out a way to improve the innovation.
2. As an invasive surgery, the authors did not mention any painless treatment after the animal surgery in the method part. It does not meet animal welfare requirements.
3. In Figure 5, the authors did not label the molecular weight of LTCC and GAPDH.
In Figure 5, according to the GAPDH bands, the phosphorylated PLB and total PLB were from different gels, which may cause an error in the result.
Author Response
July 19th, 2023
International Journal of Molecular Sciences
Ref: 2459223
Dear Editor Pongsakorn Vithayanon,
We are very grateful for the attention given to our study. The valuable comments and suggestions made by the reviewer allowed us to clearly improve the manuscript. A rebuttal letter addressing all the questions raised by the referee is being sent. Modifications made in the text and graphs to meet the reviewer's suggestions are all highlighted in yellow in the revised manuscript. The lines in the text where those modifications were inserted are also indicated.
We hope that in its thoroughly revised form, the manuscript can be considered for publication in the International Journal of Molecular Sciences. All the authors are aware and agree with the content of the manuscript and declare no conflict of interest. In submitting this work for publication, we attest that the manuscript or its essence has not been published, in whole or in part, and will not be submitted for publication elsewhere until the editors of International Journal of Molecular Sciences have made a final decision regarding its acceptability.
Thank you for your time and consideration.
Sincerely, on behalf of all co-authors,
Antonio Carlos Cicogna, MD, PhD
Department of Internal Medicine
Botucatu Medical School
São Paulo State University - Unesp
Botucatu, São Paulo, Zip code: 18618687, Brazil
Lab Phone: +55 14 3880 1611
E-mail: ac.cicogna@unesp.br
Point-by-point answers of reviewer’s comments
Thank you very much for carefully reviewing our study. We appreciate your time and effort in helping us improve our manuscript. We transcribe below all the points questioned and answer each one of them. Modifications made in the text and graphs are all highlighted in yellow in the revised manuscript. The lines in the text where those modifications were inserted are also indicated.
Reviewer 1
- The innovation of this manuscript is poor. The benefit of aerobic excise training on heart failure treatment and cardiac ca2+ handling has been widely reported in the past 20 years. Indeed, most of those studies used MI or other types of cardiac hypertrophy models, not for aortic stenosis, but the innovation of this study still needs to be improved. Authors have to figure out a way to improve the innovation.
Response: It is very important what the esteemed reviewer considers. However, we need to point out that even though the training effect of aerobic exercise on a diseased heart has been extensively studied over the last 20 years, knowledge about the adequacy of important molecular mechanisms in this context is still insufficient and divergent, especially in models of pressure overload. serious.
It is always important to emphasize that there is still a strong paradigm regarding the use of aerobic exercise training as a therapeutic support in patients with aortic stenosis. In this disease, surgery is always indicated and exercise is generally contraindicated. In the literature, few studies address this issue (1,2), especially thinking about younger patients. These papers allege that no guidelines have been established for rehabilitation in patients with severe AS, and exercise loading was contraindicated for this population
It is evident that physical exercise will not influence the aggressor agent, which causes pressure overload (in the case of our experimental model, it would be the clip, and in the human, the aortic valve). However, our initial studies and also those of other groups in the last 10 years have shown that, regardless of the cure of the aggressor agent, exercise is capable of improving and qualifying the heart muscle, as well as bringing about substantial peripheral improvements, of which we can mention the improvement of the lungs (5), skeletal muscles (3,6), kidneys and liver (data not yet published). Our experiments have shown that these positive adaptations have a great relevance in the animals' quality of life/functional capacity and longevity. Our research (3-8) can be pioneers and references for the beginning of the study of aerobic exercise training protocols for patients with aortic stenosis and mild, moderate or severe cardiac dysfunction or also with heart failure.
Aortic stenosis and the experimental model of aortic stenosis and its systemic repercussions (3-8) are completely different from other diseases and from other experimental models of cardiac aggression, including acute myocardial infarction and spontaneously hypertensive rats. It is very complicated for the scientific and medical community to seek grounding in other experimental models to have theoretical support and start testing aerobic exercise protocols in people with aortic stenosis.
We believe that our latest published articles are of paramount importance for understanding that physical exercise has therapeutic potential for patients with aortic stenosis at different levels of severity of cardiac dysfunction, as well as for understanding molecular, cellular and cardiac pump adaptations, which differ from the responses observed in other cardiac diseases.
1- Arai H, Nozoe M, Matsumoto S, et al. Exercise Training for Patients With Severe Aortic Stenosis in a Convalescent Rehabilitation Ward ― A Retrospective Cohort Study. Circ Rep. 9; 3(7): 361–367, 2021.
2- Maréchaux S, Ennezat PV, Le Jemtel TH, et al. Left ventricular response to exercise in aortic stenosis: an exercise echocardiographic study. Echocardiography. 24(9):955-9, 2007.
3- Gomes MJ, Martinez PF, Campos DH, et al. Beneficial Effects of Physical Exercise on Functional Capacity and Skeletal Muscle Oxidative Stress in Rats with Aortic Stenosis-Induced Heart Failure. Oxid Med Cell Longev. 2016.
4- Souza, RWA, Fernandez GJ, Cunha JPQ, et al. Am. J. Physiol. Heart. Circ. Physiol. 309, H1629–H1641, 2015.
5- de Souza PAT, de Souza RWA, Soares LC, et al. Aerobic training attenuates nicotinic acethylcholine receptor changes in the diaphragm muscle during heart failure. Histol. Histopathol. 30, 801–11, 2015.
6- Souza RWA, Piedade WP, Soares LC, et al. Aerobic exercise training prevents heart failure-induced skeletal muscle atrophy by anti-catabolic, but not anabolic actions. PloS one. 9, 2014.
7- Mota GAF, de Souza SLB, da Silva VLS, et al. Cardioprotection Generated by Aerobic Exercise Training is Not Related to the Proliferation of Cardiomyocytes and Angiotensin-(1-7) Levels in the Hearts of Rats with Supravalvar Aortic Stenosis. Cell. Physiol. Biochem. 54, 719-35, 2020.
8- de Souza SLB, Mota GAF, da Silva VLS, et al. Adjustments in β-Adrenergic Signaling Contribute to the Amelioration of Cardiac Dysfunction by Exercise Training in Supravalvular Aortic Stenosis. Cell. Physiol. Biochem. 54, 665-81, 2020.
- As an invasive surgery, the authors did not mention any painless treatment after the animal surgery in the method part. It does not meet animal welfare requirements.
Reponse: In fact, information about post-surgical treatment is very important in this type of study. We have included the excerpt about treatment in the "Aortic stenosis surgery", section “Materials and Methods”. Thanks to the reviewer for the note.
- In Figure 5, the authors did not label the molecular weight of LTCC and GAPDH.
Response: We are grateful for the thorough correction by the reviewer and we inform you that we have already changed what was exposed in the text.
In Figure 5, according to the GAPDH bands, the phosphorylated PLB and total PLB were from different gels, which may cause an error in the result.
Response: Thanks to the reviewer for their consideration. However, our group has historically performed the analysis of PLB and phosphorylated PLB in different gels (1-4), and this was never pointed out to us as a methodological error. If the reviewer wants to propose another analysis dynamic, we are open to adapting it.
1- De Tomasi LC, Campos DHS, Sant’Ana PG, et al. Pathological hypertrophy and cardiac dysfunction are linked to aberrante endogenous unsaturated fatty acid metabolism. PLoS One. 13: e0193553, 2018.
2- Deus AF, da Silva VL, Souza SLB, et al. Myocardial Dysfunction after Severe Food Restriction Is Linked to Changes in the Calcium-Handling Properties in Rats. Nutrients. 11(9):1985, 2019.
3- Mazeto IFS, Okoshi K, Silveira CFSMP, et al. Calcium homeostasis behavior and cardiac function on left ventricular remodeling by pressure overload. Braz J Med Biol Res. 54(4): e10138, 2021.
4- da Silva VL, Souza SLB, Mota GAF, et al. The Dysfunctional Scenario of the Major Components Responsible for Myocardial Calcium Balance in Heart Failure Induced by Aortic Stenosis. Arq Bras Cardiol. 118(2):463-475, 2022.

Reviewer 2 Report
In this manuscript da Silva et al have studied the effects of aerobic exercise training on cardiac function and calcium handling in a rat model of heart failure. The manuscript includes various relevant methods for studying cardiac function. However, serious issues exist: some conclusions appear hastily drawn, and small additional methodology would greatly deepen the mechanistic understanding of the aberrations in calcium handling and their role in the impaired cardiac function observed in AoS rats. Calcium handling issues need improvement here.
Major issues:
The abstract lacks presentation of robust results and concentrates more on conclusions. Eg. after describing the methodology the conclusion is presented: “AET attenuated the diastolic dysfunction and benefited the systolic function”. Results that these conclusions are based on should be presented first.
What is the p-value of LVDD in the AoS-ET rats before vs after ET? LVDD seems to have grown similarly to the AoS group.
Most important point: Page 6: SR calcium content should be evaluated and quantified in isolated ventricular cardiomyocytes with depletion of the SR calcium stores with caffeine. So more experimental data required. This simple method would help to answer your question eg. on the lack of inotropic response in AoS rats. Most of the current text in the discussion is around this issue, that is easily experimented on in the isolated ventricular cardiomyocytes.
What is the calcium transient duration (eg. from 10% rise to 90% decay time) in all the groups?
Figure 4: how would TCD 80? or 90% look like? Even greater prolongation in the AoS decay time? Implying decreased SERCA2a function...
Please show representative examples of calcium transients from the different groups.
Did you observe diastolic calcium release events in calcium imaging on isolated cardiomyocytes, calcium sparks? These would be expected with the increased SERCA2a and NCX expression levels that you show in AoS rats. If not, this needs to be mentioned too.
Are all the cardiomyocytes ventricular? Where in the heart are the isolated cardiomyocytes from? Methods should clearly explain this important issue, not only as a reference.
I find no mention of arrhythmias or mortality. These should be stated.
Figure 5: looking at the bars on the left and the gels on the right, one gets confused about the bar graph results, eg. looking at AoS group in B and F panels, the corresponding gels blots seem contradictory; RyR seems at least as strong as SERCA, and stronger than the other groups, opposite to what the bar graphs show. Please check and confirm this is as presented in the bar graphs.
How do qPCR results for these important players in calcium handling look like? So more experimental data required.
Page 16 calcium imaging methods should state the frame rate of the imaging recordings, and the software used for analysis.
Generally, heart failure is associated with the following changes in expression and activity of these important calcium handling proteins: LTCC no big change, RyR2 most often decreased, SERCA2a decreased (this is a fairly consistent finding), PLB increased, NCX increased.
In your findings, the most strikingly different to these, is the increased expression of SERCA2a in AoS rats. However, you find the calcium transient decay time to be increased, suggesting decreased SERCA2a activity. These two findings are contradictory, and need to be explored (SR calcium stores quantified with caffeine, as mentioned above) and discussed.
Furthermore, in the abstract you conclude that “AET increased SERCA2a activity”, although you show that in AoS-ET rats the calcium transient decay time did NOT decrease, and you show that SERCA2a expression decreased. How have you made this conclusion? Explain that in light of your findings that I mention here?
Some minor issues:
Line 65: “this positive adequacy is related to the attenuation of the homeostatic damages of Ca2+ handling”. Unclear and confusing terms especially for the unfamiliar reader: positive adequacy and homeostatic damages of calcium handling. These need to be elaborated and clarified to make the text more readable.
line 338 “capacidade funcional”. English should be used throughout.
page 5, line 17: “functionality in exercised cardiac animals”. This is an example of for the need of language revision: functionality vs function. Cardiac animals?
Still another example from line 39 of page 5: “after 28 weeks of surgery” I assume is meant to be: 28 weeks after surgery.
Another example is the use of singular vs plural, eg, “protein” instead of proteins in the subtitle on page 16, line 140.
page 5, line 27: again the words “positive adequacy” These should be replaced with the appropriate term, here eg. “increase”? And in the above example of line 65 removing those words altogether.
page 6, line 37: “these” should be replaced by eg “an inotropic” to make the text more readable and understandable.
page 6 first paragraph is mainly discussion, which should take place later.
page 8, line 107: “presented alterations” is vague and doesn’t tell the reader anything. Elaborate: increased, decreased, improved etc.
page 10, line 144: define abbreviations at first mention, here “LT”, and then use consistently, eg. line 150 on the same page.
Discussion line 177; a general statement as “metabolic profile” should not be used when a defined variable such as lactate threshold is meant.
line 178 what is meant by “enhance…signs of heart failure”? The current text means that signs of heart failure are strengthened.
line 179 “recapture potential” is an unnecessarily complicated way to say “reuptake”.
line 185 the term “adequacy” is misused here again. Better to say “remodeling” or just “change”.
line 188 “exacerbation of the extracellular matrix,” re-word this eg. “disorganization”.
line 195 “RV” define abbreviations at first mention, and use only if that term is used several times in the text. Check this for all abbreviations used.
line 204 replace “aggression”.
line 237, TFa?
In the above text.
Author Response
July 19th, 2023
International Journal of Molecular Sciences
Ref: 2459223
Dear Editor Pongsakorn Vithayanon,
We are very grateful for the attention given to our study. The valuable comments and suggestions made by the reviewer allowed us to clearly improve the manuscript. A rebuttal letter addressing all the questions raised by the referee is being sent. Modifications made in the text and graphs to meet the reviewer's suggestions are all highlighted in yellow in the revised manuscript. The lines in the text where those modifications were inserted are also indicated.
We hope that in its thoroughly revised form, the manuscript can be considered for publication in the International Journal of Molecular Sciences. All the authors are aware and agree with the content of the manuscript and declare no conflict of interest. In submitting this work for publication, we attest that the manuscript or its essence has not been published, in whole or in part, and will not be submitted for publication elsewhere until the editors of International Journal of Molecular Sciences have made a final decision regarding its acceptability.
Thank you for your time and consideration.
Sincerely, on behalf of all co-authors,
Antonio Carlos Cicogna, MD, PhD
Department of Internal Medicine
Botucatu Medical School
São Paulo State University - Unesp
Botucatu, São Paulo, Zip code: 18618687, Brazil
Lab Phone: +55 14 3880 1611
E-mail: ac.cicogna@unesp.br
Point-by-point answers of reviewer’s comments
Thank you very much for carefully reviewing our study. We appreciate your time and effort in helping us improve our manuscript. We transcribe below all the points questioned and answer each one of them. Modifications made in the text and graphs are all highlighted in yellow in the revised manuscript. The lines in the text where those modifications were inserted are also indicated.
Reviewer 2
Major issues:
1- The abstract lacks presentation of robust results and concentrates more on conclusions. Eg. after describing the methodology the conclusion is presented: “AET attenuated the diastolic dysfunction and benefited the systolic function”. Results that these conclusions are based on should be presented first.
Response: Dear reviewer, we agree with the proposal and thank you for your consideration. The Abstract really should contain more detailed information, especially the results of the study. However, there is a character limit for the "Abstract" Section, and therefore, unfortunately, we were unable to describe in detail each variable that presented adaptation by the exercise. We created a new Abstract at your request. However, it seemed impracticable to us for publication in the Journal.
Abstract: Aerobic exercise training (AET) has been used in the management of heart disease. AET may, totally or partially, restore the activity and/or expression of proteins that regulate calcium (Ca2+) handling, optimize intracellular Ca2+ flow, and attenuate cardiac functional impairment in failing hearts. However, the literature presents conflicting data regarding the effects of AET on Ca2+ transit and cardiac function in rats with heart failure resulting from aortic stenosis. The objective of this study was to evaluate the effects of AET on calcium handling and cardiac function in rats with heart failure due to aortic stenosis. Wistar rats were distributed into two groups: control (Sham; n = 61) and aortic stenosis (AoS; n = 44). After 18 weeks, the groups were redistributed into: non-exposed to exercise training (Sham, n = 28 and AoS, n = 22) and trained (Sham-ET, n = 33 and AoS-ET, n = 22) for 10 weeks. Treadmill exercise training was performed with a velocity equivalent to the lactate threshold. Echocardiogram, isolated papillary muscle, and isolated cardiomyocyte analyzed cardiac function. During isolated papillary muscle assay and isolated cardimyocyte was evaluated Ca2+. The expression of regulatory proteins of diastolic Ca2+ was analyzed via Western Blot. The following results of improvement in cardiac mechanics by AET were observed by the echocardiogram and isolated papillary muscle: the attenuation of the diastolic dysfunction was expressed by the increase in the mitral velocity annulus during atrial contraction and e-wave deceleration time and reduction of the e-wave, of the ratio between left atrium and aortic diameter, the ratio between filling flow peak and mitral annulus velocity during early ventricular filling and rest tension, while the benefits of the systolic function presented by the variables posterior wall systolic velocity and ejection fraction.. Moreover, AoS-ET animals presented better response to post-rest contraction, and SERCA2a and L-type Ca2+ channels blocked than the AoS. Furthermore, AET was able to improve aspects of the mechanical function and the responsiveness of the myofilaments to the Ca2+ of the AoS-ET animals. AoS animals presented alteration in the protein expression of the SERCA2a and NCX, and AET restored SERCA2a and NCX levels near normal values. Therefore, AET increased SERCA2a activity, improved the cellular Ca2+ influx mechanism, and increased myofilament responsiveness to Ca2+, attenuating cardiac dysfunction at cellular, tissue, and chamber levels in animals with aortic stenosis and heart failure.
2- What is the p-value of LVDD in the AoS-ET rats before vs after ET? LVDD seems to have grown similarly to the AoS group.
Response: Dear reviewer, thank you for your question. It is noteworthy that we also understand the variable left ventricular diastolic diameter (LVDD) as an important marker of pathological remodeling. The outcome we found shows that the heart remains diseased even after physical training, because, among other findings, there was no difference in LVDD between the AoS-ET and AoS groups 10 weeks after the beginning of the exercise. However, we showed through other echocardiographic variables and experimental techniques that the heart of AoS-ET animals has better functionality.
In fact, looking at the data, the percentage change in LVDD between weeks 18 and 28 seems similar when comparing the AS and AS-ET groups. In this study, we did not carry out this statistical analysis of the "before and after" for any variable and therefore we do not have the p value of the aforementioned comparison.
3- Most important point: Page 6: SR calcium content should be evaluated and quantified in isolated ventricular cardiomyocytes with depletion of the SR calcium stores with caffeine. So more experimental data required. This simple method would help to answer your question eg. on the lack of inotropic response in AoS rats. Most of the current text in the discussion is around this issue, that is easily experimented on in the isolated ventricular cardiomyocytes.
Response: Dear Reviewer, we appreciate the commentary and thank you very much for your consideration. We totally agree that SR calcium content could have been evaluated and quantified in isolated ventricular cardiomyocytes with depletion of the SR calcium stores with use of caffeine. In this sense, we know that cardiac sarcoplasmic reticulum (SR) serves as a Ca2+ reservoir for contraction, which reuptakes intracellular Ca2+ during relaxation. The SR Ca2+ reserve available for beats is determinate for cardiac contractibility, and the removal of intracellular Ca2+ is critical for cardiac diastolic function. In addition, we already had knowledge about the use of caffeine for inducing total SR Ca2+ release, but when the project was designed, we only had the perspective of performing the analysis of SR storage and release capacity from post-rest contraction (PRC) in isolated papillary muscle, since we were standardizing the isolated cardiomyocyte technique and pharmacological assays. It should be noted that we are unable to perform this requested analysis in this moment because we have no animals available with experimental protocols similar to the current one. but it would really enrich the findings and discussions of the article. We thank you again for your suggestion, which will be applied in future studies.
4- What is the calcium transient duration (eg. from 10% rise to 90% decay time) in all the groups?
Figure 4: how would TCD 80? or 90% look like? Even greater prolongation in the AoS decay time? Implying decreased SERCA2a function...
Please show representative examples of calcium transients from the different groups.
Did you observe diastolic calcium release events in calcium imaging on isolated cardiomyocytes, calcium sparks? These would be expected with the increased SERCA2a and NCX expression levels that you show in AoS rats. If not, this needs to be mentioned too.
Response: Dear Reviewer, we appreciate the commentary and thank you very much for your consideration. The explanation is as follows: we would like to analyze the Intracellular Ca2+ transient from Ca2+ amplitude, diastolic and systolic Ca2+, time to Ca2+ peak, and time to 50% Ca2+ peak and decay. Thus, when we use a TCD50% means a calcium transient duration at 50% of decay following the peak amplitude, which is reported to have a high correlation with the ratio of peak decay time to peak rise time (D/R ratio). Decay of the calcium transient has been shown to follow an exponential time course, and the rate of decay can be correlated with the expression of the sarco-(endo)plasmic reticulum calcium-ATPase pump SERCA2a.When calcium transient alternans is recorded with fluorescentindicators, beat-to-beat alternation in the end-diastolic level ofthe transient isusually observed. Therefore, our findings confirm this relationship, since the increase in TD50 was correlated with higher expression of SERCA2a in a aortic stenosis. In this sense, this elevation in Serca expression may be related to an attempt to recapture calcium more quickly and remove cytosolic calcium with consequent improvement in diastole, however, this was not observed, and an impaired diastole evidenced by a greater TCD50%. In addition, it should be noted that we did not make calcium sparks, because we used myocytes that were loaded with 1.0 μM Fura2-acetoxymethyl (AM) ester (Molecular Probes, Eugene, OR, USA) for calcium transient analysis, which does not provide images, only records. Thus, the emitted records were transformed into graphs as demonstrated in Figure 4.
5- Are all the cardiomyocytes ventricular? Where in the heart are the isolated cardiomyocytes from? Methods should clearly explain this important issue, not only as a reference.
Response: Dear Reviewer, we appreciate the commentary and totally agree this information is missing the text. As requested, the new methodology is described below:
Under anesthesia, rats from each group were euthanized and the hearts were quickly removed by median thoracotomy and enzymatically isolated as previously described [27]. Briefly, the hearts were cannulated and retrograde perfusion of the aorta was performed in Langendorff system (37oC) with a modified isolation digestion buffer solution (DB), a calcium-free solution containing 0.1 mM ethylene glycol-bis (ß-aminoethyl ether)-N, N, N’, N’- tetraacetic acid (EGTA) and N-[2-hydro-ethyl]-piperazine-N’-[2-ethanesulfonic acid] (HEPES) equilibrated with 5% CO2-95% O2 for ~3 to 5 min. The composition of DB solution was (mM): 130 NaCl, 1.4 MgCl2, 5.4 KCl, 25 HEPES, 22 glucose, 0.33 NAH2PO4, and pH 7.39. Afterwards, the hearts were perfused for 20-30 minutes with a DB solution containing 1 mg/ml collagenase type II (Worthington Biochemical Corporation, UK) and Ca2+ (1 mM). Subsequently, isolated cells were placed in an experimental chamber with a glass coverslip base mounted on the stage of an inverted microscope (MyoCam-S, IonOptix, Milton, MA, USA) edge detection system with a 40× objective lens (Nikon Eclipse – TS100, USA). After the digestion process, the supernatant was removed, and the myocytes were resuspended in Tyrode’s buffer containing (in mM): 140 NaCl, 10 HEPES, 0.33 NaH2PO4; 1 MgCl2, 5 KCl, 1.8 CaCl2, 10 glucose. Cells were immersed in Tyrode’s solution, and field stimulated at 1 Hz (20 V, 5 ms duration square pulses). Cell shortening in response to electrical stimulation was measured with a video-edge detection system at a 240-Hz frame rate (Ionwizard, Ion Optix, Milton, MA, USA) and the contractile parameters were evaluated. Sarcomere length, fractional shortening (expressed as a percentage of resting cell length), maximum shortening velocity, maximum relaxation velocity, and time to 50% shortening (time to 50% peak) and 50% relaxation (time for 50% relaxation) were measured in 6 cells per animal in each experimental group.
6- I find no mention of arrhythmias or mortality. These should be stated.
Response: We appreciate the reviewer's consideration. In this study, we did not assess arrhythmias. Regarding mortality, we apologize for not having included the data initially. There was no difference in mortality between the AoS and AoS-ET groups. We inform you that we have added the following excerpt to the text:
"Furthermore, survival rate did not differ between the AoS and AoS-ET groups (data not shown)."
7- Figure 5: looking at the bars on the left and the gels on the right, one gets confused about the bar graph results, eg. looking at AoS group in B and F panels, the corresponding gels blots seem contradictory; RyR seems at least as strong as SERCA, and stronger than the other groups, opposite to what the bar graphs show. Please check and confirm this is as presented in the bar graphs.
Response: Dear reviewer, we understand your consideration and thank you. We have reviewed the blots to answer you, and we have concluded that the bars are representative. It is important to emphasize that there is another part of the blot, there are 7 animals per group and we chose to send the blots that were clearer to the article.
8- How do qPCR results for these important players in calcium handling look like? So more experimental data required.
Response: Dear reviewer, we agree with your consideration. However, as you noted, we performed many experimental techniques in our study and, unfortunately, we did not have enough financial support or time to perform the qPCR technique. We believe that the Western Blot is an important and relevant technique to show the response that we would like to analyze.
9- Page 16 calcium imaging methods should state the frame rate of the imaging recordings, and the software used for analysis.
Response: The analysis of contractility parameters was performed using an available software program (IonWizard Ionoptix, Ion Optix Contractility Systems, MyoCam-S). The MyoCam-S™ is an all-digital, variable frame rate camera that utilizes the USB 2.0 standard to remove the restrictions of analog video formats and frame grabbers. Variable frame rates (lines): 97Hz (245 lines), 250Hz (87 lines), 500Hz (36 lines), 1000Hz (10 lines). Due to differences in cell contractility speed within the different cell types, a camera should have a frame rate ranging from 25 to 200 fps (1). Thus, different cell types may require specific microscope settings, and to achieve the best contrast, some adjustments to diaphragm aperture, focus, and light intensity may be necessary. As recommended by Ion Optix company, we used a maximum frame rate of 240 fps in adult ventricular myocytes for contractility recording settings. The information is described below:
Subsequently, isolated cells were placed in an experimental chamber with a glass coverslip base mounted on the stage of an inverted microscope (MyoCam-S, IonOptix, Milton, MA, USA) edge detection system with a 40× objective lens (Nikon Eclipse – TS100, USA). After the digestion process, the supernatant was removed, and the myocytes were resuspended in Tyrode’s buffer containing (in mM): 140 NaCl, 10 HEPES, 0.33 NaH2PO4; 1 MgCl2, 5 KCl, 1.8 CaCl2, 10 glucose. Cells were immersed in Tyrode’s solution, and field stimulated at 1 Hz (20 V, 5 ms duration square pulses). Cell shortening in response to electrical stimulation was measured with a video-edge detection system at a 240-Hz frame rate (Ionwizard, Ion Optix, Milton, MA, USA) and the contractile parameters were evaluated.
1- Scalzo S., Mendonça C.A.T.F, Kushmerick C., Agero U., Guatimosim S. Microscopy-based cellular contractility assay for adult, neonatal, and hiPSC cardiomyocytes. STAR Protoc. 2022 Mar 18; 3(1): 101144.
10- Generally, heart failure is associated with the following changes in expression and activity of these important calcium handling proteins: LTCC no big change, RyR2 most often decreased, SERCA2a decreased (this is a fairly consistent finding), PLB increased, NCX increased.
In your findings, the most strikingly different to these, is the increased expression of SERCA2a in AoS rats. However, you find the calcium transient decay time to be increased, suggesting decreased SERCA2a activity. These two findings are contradictory, and need to be explored (SR calcium stores quantified with caffeine, as mentioned above) and discussed.
Furthermore, in the abstract you conclude that “AET increased SERCA2a activity”, although you show that in AoS-ET rats the calcium transient decay time did NOT decrease, and you show that SERCA2a expression decreased. How have you made this conclusion? Explain that in light of your findings that I mention here?
Response: Dear reviewer, thank you for your comment. We emphasize that when we obtained this finding we were also surprised, exactly because in the literature the outcome found is different. However, we present in the study what we checked in a triplicate experiment and we have full confidence in this result.
It is worth mentioning again that our experimental model is completely different from any other studied in the literature. Therefore, we may not base a result on a model of myocardial infarction or pressure overload in spontaneously hypertensive rats.
In an experimental model similar to ours, Hiemstra et al, 2018 analyzed the effect of chronic low-intensity exercise on the contractile dysfunction of cardiomyocytes and the damage of the adrenergic system in mini-swine with heart disease consequent to aortic stenosis. In this study, the authors evaluated calcium handling proteins and did not observe an increase in SERCA2a/Calsequestrin by exercise training, as cited by the reviewer; there was no difference for this protein between groups. Observing the figure (Figure 6A from the study by Hiemstra et al., 2018), there is a tendency to decrease the protein expression of the protein cited by exercise training. The authors did not discuss this finding.
We understand the increase in SERCA2a protein expression in the AoS group, as a cellular compensatory mechanism in an attempt to balance the calcium flow, especially during cardiomyocyte relaxation, in calcium reuptake. However, in our study this response proved to be ineffective, because the diastole remained impaired. Our findings show that the SERCA2a protein expression of the Aos-ET group is similar to the Sham group, which does not have heart disease.
The difference in SERCA2a functionality between AoS and AoS-ET is very clear in the post-rest contraction maneuver of the isolated papillary muscle technique. The AoS-ET group has a significantly greater potential response of SERCA2a functionality than the AoS group, which has practically no response (Figures 1A-C). The inotropic response of this maneuver largely depends on the ability of SERCA2a to reuptake calcium.
In the study, we bring all this discussion in the light of our findings.
Part of the Discussion:
“Studies have shown, in different experimental models, that Ca2+ handling adjustments are essential for enhancing the heart’s performance by the AET in heart failure and dysfunction [14-25]. According to Kim et al. [45], the restoration of protein expression levels related to the excitation-contraction-relaxation coupling close to the normal heart is among the positive adaptations to the therapies implemented during heart failure. Our results confirm the proposition since AoS-ET animals showed lower protein expression of SERCA2a and NCX compared to AoS, and these values were similar to the control groups; however, the data are in disagreement with the literature, which mostly points to maintenance or increase in the SERCA2a expression by AET in normal, infarcted and hypertensive animals [16,18,25,53]. In pigs with AoS and preserved ejection fraction submitted to the exercise, authors observed raised SERCA2a/PLB ratio, serine 16 phosphorylated PLB, and NCX, in addition to decreased diastolic Ca2+, which were associated with functional improvements of cardiomyocytes compared to sedentary animals [14]. However, van Deel et al. [26] did not identify improvement in cardiac function and positive adjustments in Ca2+ handling by voluntary training in mice with AoS, possibly due to the model and severity of AoS and exercise protocol. It is noteworthy that, in our study, the reduced SERCA2a in trained AoS animals was accompanied by an improvement in cardiomyocyte function in relation to AoS animals; moreover, these rats expressed beneficial values for RMV and TR50% than the sedentary animals in the analysis of isolated cardiomyocytes. In addition, the AoS-ET group improved SMV and shortening fraction, which may be related to the higher sensitivity of myofilaments to Ca2+, proposed by the improvement in the ratios of systolic Ca2+/Fractional shortening and systolic Ca2+/SMV. In agreement, previous studies have shown increased sensitivity of myofilaments to Ca2+ by exercise training in normal [54,55] and infarcted [25] rats.
The maneuvers performed during the isolated papillary muscle analysis provided support regarding the effects of AET on SERCA2a and L-channel activity in AoS animals. In the blockage of the ATP binding site of SERCA2a by cyclopiazonic acid, there was no statistical difference between AoS-ET and AoS groups for all variables and moments studied. As there was a significant decrease in SERCA2a expression in the sarcoplasmic reticulum from AoS-ET mice in relation to AoS group, the similarity of response suggests an enhanced function of SERCA2a in the AoS-ET group compared to AoS; this group should have presented better performance after the electrical stimulus pauses and a lower percentage of inhibition after ACP, due to the increased amount of SERCA2a. Authors have shown augmented SERCA2a activity promoted by interval aerobic exercise in normal [53,56] and infarcted [25] rats. As previously mentioned, this protein is an ATPase, which depends on cytosolic ATP levels to perform its normal function [57]; thus, the number and functioning of mitochondria play a fundamental regulatory role [58]. The literature shows that AET improves mitochondrial function, which is impaired in heart diseases [59], as well as decreases oxidative stress [60], raised in pathological conditions [38]. Therefore, in the present study, these mechanisms may have been related to the attenuation of the SERCA2a functional impairments.”
1- Hiemstra, J.A.; Veteto, A.B.; Lambert, M.D.; Olver, T.D.; Ferguson, B.S.; McDonald, K.S. et al. Chronic low-intensity exercise attenuates cardiomyocyte contractile dysfunction and impaired adrenergic responsiveness in aortic-banded mini-swine. J. Appl. Physiol. (1985). 2018,124, 1034–44.
Some minor issues:
11- Line 65: “this positive adequacy is related to the attenuation of the homeostatic damages of Ca2+ handling”. Unclear and confusing terms especially for the unfamiliar reader: positive adequacy and homeostatic damages of calcium handling. These need to be elaborated and clarified to make the text more readable.
12- line 338 “capacidade funcional”. English should be used throughout.
13- page 5, line 17: “functionality in exercised cardiac animals”. This is an example of for the need of language revision: functionality vs function. Cardiac animals?
14- Still another example from line 39 of page 5: “after 28 weeks of surgery” I assume is meant to be: 28 weeks after surgery.
15- Another example is the use of singular vs plural, eg, “protein” instead of proteins in the subtitle on page 16, line 140.
16- page 5, line 27: again the words “positive adequacy” These should be replaced with the appropriate term, here eg. “increase”? And in the above example of line 65 removing those words altogether.
17- page 6, line 37: “these” should be replaced by eg “an inotropic” to make the text more readable and understandable.
18- page 6 first paragraph is mainly discussion, which should take place later.
19- page 8, line 107: “presented alterations” is vague and doesn’t tell the reader anything. Elaborate: increased, decreased, improved etc
20- page 10, line 144: define abbreviations at first mention, here “LT”, and then use consistently, eg. line 150 on the same page.
21- Discussion line 177; a general statement as “metabolic profile” should not be used when a defined variable such as lactate threshold is meant.
22- line 178 what is meant by “enhance…signs of heart failure”? The current text means that signs of heart failure are strengthened.
23- line 179 “recapture potential” is an unnecessarily complicated way to say “reuptake”.
24- line 185 the term “adequacy” is misused here again. Better to say “remodeling” or just “change”.
25- line 188 “exacerbation of the extracellular matrix,” re-word this eg. “disorganization”.
26- line 195 “RV” define abbreviations at first mention, and use only if that term is used several times in the text. Check this for all abbreviations used.
27- line 204 replace “aggression”.
28- line 237, TFa?
Response: Dear reviewer, thank you very much for your thorough and thoughtful review. We believe it greatly enriched the work. All minor points have been corrected in the text.
